# Spatial mapping and determinants of intimate partner violence among married women in Sierra Leone: A cross-sectional study

Augustus Osborne[1]*, Umaru Sesay[2,3], Camilla Bangura[4], Lovel Fornah[5]

1 Institute for Development, Freetown, Western area, Sierra Leone, 2 Sierra Leone Field Epidemiology Training Program, National Public Health Agency, Western Area, Freetown, Sierra Leone, 3 Africa Field Epidemiology Network, Western Area, Freetown, Sierra Leone, 4 Department of Biological Sciences, School of Basic Sciences, Njala University, PMB, Freetown, Sierra Leone, 5 Department of Public Health, Ernest Bai Koroma University of Science and Technology, Makeni Campus, Makeni, Sierra Leone

* augustusosborne2@gmail.com

## Abstract

### Background

Intimate partner violence is a pervasive public health and human rights issue, dispro-portionately affecting women worldwide. In Sierra Leone, where gender inequalities and socio-cultural norms remain entrenched, intimate partner violence is a signif-icant concern, with severe consequences for women's physical, emotional, and social well-being. Understanding the spatial distribution and determinants of intimate partner violence is crucial for designing targeted interventions to address this issue. Using data from the 2019 Sierra Leone demographic and health survey, this study explored the geographic patterns of intimate partner violence and identified key socio-demographic and behavioural factors associated with its prevalence among married women.

### Methods

The study employed data from the 2019 Sierra Leone demographic and health sur-vey. The study comprised of 3,611 married women between the ages of 15 and 24. Spatial autocorrelation and Moran's I statistic were employed to analyse the spatial distribution of intimate partner violence. The study utilised mixed-effect multilevel binary logistic regression using a four-model framework to determine the factors related to intimate partner violence. The findings were presented as adjusted odds ratios (aOR) and a 95% confidence interval (CI).

### Results

The study revealed an overall intimate partner violence prevalence of 56%, with physical violence accounting for 38.2%, sexual violence for 6.2%, and emotional

**Data availability statement:** The data used for this study is freely available at https://dhspro-gram.com/data/dataset/Sierra-Leone_Standard-DHS_2019.cfm?flag=0

**Funding:** The author(s) received no specific funding for this work.

**Competing interests:** The authors have declared that no competing interests exist.

**Abbreviations:** aOR, Adjusted Odds Ratio; CI, Confidence Interval; DHS, Demographic and Health Survey; IPV, Intimate Partner Violence; MEASURE DHS, Monitoring and Evaluation to Assess and Use Results Demographic and Health Surveys; STROBE, Strengthening the Reporting of Observational Studies in Epidemiology

violence for 45.9% among married women in Sierra Leone. Hotspot districts for intimate partner violence were identified in the Western area (urban and rural areas) and the Northwestern province (Kambia and Karene). At the same time, Bo, Kenema, and Bombali, the provincial headquarters of the Northern, Eastern, and Southern provinces, were found as cold spot districts for intimate partner violence. Factors associated with intimate partner violence included married women aged 25–29, those with one-two partner controlling behaviour, and those who provided one-two justifications for wife beating. Furthermore, married women exposed to interparental violence and those who resided in the Northwestern, Northern, and Western area had a higher likelihood of experiencing intimate partner violence.

## Conclusion

The high prevalence of intimate partner violence, particularly in hotspot districts like the Western and Northwestern province, underscore the need for province-specific interventions to protect women and reduce violence. Efforts should focus on challenging harmful cultural norms that justify wife-beating and controlling behaviours while also addressing the intergenerational cycle of violence by supporting women exposed to interparental violence. Policies must prioritise targeted education, community engagement, and enforcement of laws against intimate partner violence. Integrating intimate partner violence prevention into broader health, social, and legal systems is essential to ensure a coordinated and sustainable response to this pervasive issue.

## Introduction

Intimate Partner Violence (IPV) is any behaviour within an intimate relationship that causes physical, psychological, or sexual harm to those involved [1]. This includes acts of physical aggression, sexual coercion, psychological abuse, and controlling behaviours [1]. IPV is a pervasive public health issue driven by a complex interplay of factors, including gender inequality, cultural norms, economic dependence, and lack of legal protections [2]. Its effects are far-reaching, leading to physical injuries, mental health issues such as depression and anxiety, and even death [1]. IPV also has broader societal implications, including economic costs and intergenerational cycles of violence [3]. In Sierra Leone, where gender inequality remains deeply entrenched, and the impacts of a decade-long civil war continue to affect social structures, IPV is a critical public health and human rights issue [4]. Understanding the spatial distribution and determinants of IPV in Sierra Leone is essential for developing targeted interventions to protect women and promote gender equity.

Globally, IPV is a widespread issue, with the World Health Organization (WHO) estimating that 1 in 3 women (approximately 30%) have experienced physical or sexual violence by an intimate partner at some point in their lives [5]. The prevalence of IPV is particularly high in low- and middle-income countries, where patriarchal

norms and limited access to justice systems exacerbate the problem [5]. In sub-Saharan Africa, IPV rates are among the highest in the world, with an estimated 37% of women reporting experiences of physical or sexual violence by a partner [5]. In Sierra Leone, the statistics are even more alarming. According to the 2019 Sierra Leone Demographic and Health Survey (SLDHS), 62% of women aged 15–49 reported experiencing some form of IPV in their lifetime, with 45% reporting IPV in the 12 months preceding the survey [6]. These figures underscore the urgent need for comprehensive strategies to address IPV and its underlying causes in Sierra Leone.

In response to the high prevalence of IPV, Sierra Leone has implemented several programs and policies aimed at reducing violence against women. The Domestic Violence Act of 2007 provides a legal framework for addressing IPV, while the National Gender Strategic Plan seeks to promote gender equality and empower women [7]. Additionally, non-governmental organisations (NGOs) and community-based initiatives have been working to raise awareness about IPV, provide support services to survivors, and challenge harmful cultural norms [8]. However, these efforts face significant challenges, including weak enforcement of laws, limited access to support services in rural areas, and deeply rooted societal attitudes that normalise violence against women [8]. Furthermore, the lack of reliable data on the spatial distribution and determinants of IPV hinders the development of targeted interventions. Overcoming these challenges requires a more nuanced understanding of the issue, including how it varies across different regions and communities in Sierra Leone.

Existing studies on IPV among women have identified various factors associated with its occurrence. These include individual-level factors such as age, education, and employment status; relationship-level factors such as marital conflict and partner alcohol use; and community-level factors such as poverty, gender norms, and access to support services [9–24]. Research has also highlighted the role of structural determinants, including legal and policy frameworks, in shaping women's vulnerability to IPV [25,26]. While these studies have provided valuable insights, most have focused on national or regional prevalence rates without exploring the spatial variation in IPV or the specific contextual factors driving it. Understanding these spatial patterns is crucial for designing interventions tailored to the unique needs of different communities, particularly in countries like Sierra Leone, where geographic disparities in access to resources and services are pronounced.

In Sierra Leone, research on IPV remains limited, with most studies focusing on general gender-based violence or the broader impacts of the civil war on women's well-being [27–32]. No studies have examined the spatial distribution of IPV and the specific determinants of violence at the community level. This gap in literature is significant, as it limits the ability of policymakers and practitioners to develop targeted, evidence-based interventions. This study seeks to fill this gap by using data from the 2019 SLDHS to map the spatial distribution of IPV and identify its key determinants among women in Sierra Leone. By doing so, it provides critical insights into the geographic and contextual factors driving IPV, contributing to the existing literature on IPV and informing more effective policies and programs. The findings can improve women's lives in Sierra Leone by supporting efforts to reduce IPV and promote gender equality, ultimately contributing to the country's broader development goals.

## Methods

### Study design and sampling methods

This study employed the 2019 SLDHS. The SLDHS is a periodic cross-sectional survey that gathers data on demographic, health, and nutritional factors among reproductive-age men, women, and children. The 2019 SLDHS utilised a two-stage stratified probability cluster sampling method [6]. The stratification variable was the residential area, encompassing urban and rural regions. Enumeration areas (EAs) were selected using probability proportionate to size. During the second phase, households were chosen using systematic random selection. From May to August 2019, 578 enumerators were surveyed, and 13,399 households were questioned throughout the data collection period. Extensive information regarding the sampling approach can be found in the final SLDHS report [6]. The research included 3,611 married women between the ages of 15 and 49. This study adhered to the Strengthening Reporting of Observational Studies in Epidemiology (STROBE) guidelines [33].

## Data extraction and management

The DHS program website optimized registration and requests, making dataset access and download easy. We requested the data by clearly stating the research goals. After receiving a letter of authorization and access from (archive@dhsprogram.com), we downloaded the SLDHS 2019 dataset in STATA format from the Measure DHS program website at (http://www.dhsprogram.com). To visualize IPV prevalence hotspots and cold spots, we submitted a formal request for latitude and longitude data. The geographic coordinates were randomly altered by 5 km to protect participant confidentiality. Stata SE 18 was used to clean, recode, merge, and append the IPV-related dataset. We addressed missing data through multiple imputations for non-essential missing values and implemented complete case analysis when the amount of missing data was minimal. Furthermore, we utilized DHS sample weights to adjust for non-response, ensuring that our results were representative. Finally, we recognize that the DHS program's method for handling geographic displacements sometimes leads to missing or inaccurate location data for sensitive areas. To address this, we appended the GPS coordinates from SLDHS in the IPV-related dataset, as the DHS guidelines recommended, and then we ran tests to see how the results changed with and without the added GIS data.

## Variables

**Outcome variable.** The outcome variable, IPV among married women, is derived from the 2019 Sierra Leone Demographic and Health Survey (DHS) through responses to a series of standardised questions within the domestic violence module. This module collects self-reported data on various forms of IPV, including physical, emotional, and sexual violence, experienced by women aged 15–49 who are currently or previously married or in a union. The IPV variable is typically constructed by aggregating responses to specific questions regarding acts such as being slapped, pushed, hit, kicked, forced into sexual acts, or insulted by a partner. The responses are coded into a binary composite measure (0 = no IPV, 1 = any IPV), in line with established DHS methodology and prior studies [21–24].

We employed a consistent measure of IPV for both spatial and multilevel analyses to ensure the comparability of our findings. In the spatial analysis, we linked each of the three types of IPV-physical, emotional, and sexual violence represented as binary variables, to the GPS coordinates of DHS clusters. This approach allowed us to identify geographic patterns while accounting for spatial autocorrelation. For the multilevel analysis, we maintained the same binary outcome (IPV) but focused on differentiating between individual and community-level factors using hierarchical random effects.

**Explanatory variables.** The study utilized fourteen socio-demographic variables based on their availability in the dataset and their relevance as identified in prior research on IPV [21–24]. These variables were selected to capture a broad range of factors that influence IPV, including individual characteristics (e.g., age, educational level, employment status), partner-related factors (e.g., controlling behaviors, alcohol consumption), societal influences (e.g., exposure to family planning media, justification of spousal abuse, interparental violence), and contextual factors (e.g., wealth index, residential location, geographical region). Certain socio-demographic variables that are known to influence IPV were not included in this study due to their unavailability in the dataset. For example, variables such as mental health status, history of childhood abuse, and detailed relationship dynamics (e.g., length of marriage or partnership) were not available in the DHS dataset.

## Spatial Analysis

ArcGIS version 4.3.1 was utilised for spatial autocorrelation analysis, identification, and interpolation of intimate partner violence among married women in Sierra Leone. The analysis shapefiles were acquired from the Spatial data repository [34], and we used sub-national regional boundaries for the SLDHS, specifically we used administrative level 2 that comprises of 16 districts to maximize analytical precision while maintaining relevance for policy implementation in Sierra Leone.

A spatial autocorrelation study (Global Moran's I) was performed to determine whether IPV among married women was dispersed, clustered, or randomly distributed across districts in Sierra Leone. Moran's I statistic ranges from −1–1, with

positive values indicating clustering, negative values indicating dispersion, and values close to zero representing a random distribution of IPV rates [35]. The z-score and p-values were utilised to determine the statistical significance of spatial autocorrelation. We employed the Getis-Ord GI analysis to investigate the spatial autocorrelation variation of IPV among married women across different districts in Sierra Leone. The Getis-Ord GI metric evaluates local spatial autocorrelation, pinpointing hotspots and coldspots of intimate partner violence among married women in the area [36]. In this context, hotspots refer to districts with significantly high rates of IPV, where the prevalence is statistically higher than in surrounding areas. Conversely, cold spots refer to districts with significantly low rates of IPV, where the prevalence is statistically lower than in surrounding areas [36]. A p-value under 0.05 was considered statistically significant for evaluating clustering or dispersion.

The Empirical Bayesian Kriging (EBK) method was utilised to interpolate IPV among married women in areas lacking data, improving the understanding of spatial distribution. EBK is proficient at alleviating variations and predicting values in unsampled regions by employing neighbouring data points [37]. The Bayesian approach to spatial kriging offers significant advantages over traditional interpolation methods, such as Inverse Distance Weighting (IDW) or frequentist kriging, particularly in the context of complex health outcome data. Notably, EBK excels in mitigating variability and predicting values in regions that remain unsampled by exploiting data points from neighboring locales. This technique harnesses the principles of spatial autocorrelation to discern spatial trends within the data and to forecast the underlying dependency structures [38]. The method incorporates simulations and subsetting paradigms for parameter estimation, deriving the semivariogram from existing data points, thereby enabling the estimation of values at unobserved locations using a unified semivariogram. It is crucial to acknowledge that this approach presumes the semivariogram calculated for the interpolation zone accurately represents the true semivariogram [39].

In comparative assessments against alternative spatial interpolation techniques, bayesian kriging has demonstrated superior performance tailored to our specific research imperatives. EBK has exhibited enhanced accuracy through the integration of auxiliary variables together with spatial autocorrelation [40]. Furthermore, Bayesian Kriging addresses the tendency of Ordinary Kriging to produce overly optimistic prediction variances by conceptualizing model parameters as random variables, which facilitates more robust uncertainty quantification [41].

The implementation of Bayesian spatial Kriging has profoundly influenced our findings, establishing it as an invaluable tool for the exploration of patterns and risk factors associated with IPV. Numerous studies have successfully employed this methodology to identify high-risk areas in conjunction with pertinent neighborhood-level characteristics [42–45]. Moreover, this modeling approach adeptly identifies hotspots, regions characterized by uncertain estimates, and covariates that elucidate the spatial patterns of the outcome variable [43]. The Bayesian spatial modeling framework enabled us to account for both fixed covariate effects and spatially structured random effects, thereby facilitating a comprehensive understanding of IPV patterns and informing targeted prevention strategies [44,45]. We recognize that the application of Bayesian spatial techniques presents computational challenges and necessitates meticulous selection of prior distributions. To address these concerns, we undertook extensive sensitivity analyses to evaluate the influence of varying prior specifications on our study's findings.

**Mixed effect analysis**

Statistical analyses were conducted using Stata version 18.0 (Stata Corporation, College Station, TX, USA). We evaluated the distribution of the explanatory variables concerning the result variable and utilised a Pearson chi-square test to illustrate their relationships. A multicollinearity evaluation was performed using the variance inflation factor (VIF) before the regression analysis. The findings demonstrated that none of the variables analysed had considerable collinearity, as all values were below five. A mixed-effects multilevel binary logistic regression analysis employing four models was conducted to identify the factors associated with IPV among married women. Model I, devoid of explanatory factors, illustrated the changes in IPV among married women due to clustering at the primary sampling units (PSU), allowing us to quantify

the baseline clustering of IPV and calculate the intraclass correlation coefficient (ICC). Model II included individual-level factors. Our variable selection process was based on the preliminary bivariate analyses. Thus, we retained all individual-level variables that showed associated ($p < 0.05$) in Model II. Model III included contextual-level variables. We incorporated contextual-level variables that were statistically significant from the bivariate analyses. These variables allowed us to examine the broader community factors influencing IPV risk beyond individual factors. Model IV included all the explanatory variables that were statistically significant from the preliminary bivariate analyses.

The mixed-effects regression analysis yielded results that included both fixed and random effects. The fixed-effect analysis revealed the correlation between the explanatory variables and intimate partner violence among married women. The results were reported as an adjusted odds ratio (aOR) and their corresponding 95% confidence intervals (CI). The random effects findings underscore the variations in intimate partner violence among married women. All four models utilized the intra-cluster correlation coefficient values to evaluate the variation. All analyses were weighted using the svyset command in Stata, which integrates sample weights, various phases of clustered sampling, and stratification to accommodate the intricate peculiarities of the DHS dataset. In our research, we utilized the svyset command in Stata to provide the survey design settings, including sample weights, clustering, and stratification. This methodology is essential for obtaining unbiased estimates and accurate standard errors in analyzing complex survey data. The DHS dataset provides sample weights that reflect the selection probability at every stage of the sampling process. By employing these weights, we ensured that our estimations appropriately reflect the sample population, rectifying any disproportionate sampling of specific subgroups. This is essential in our research to accurately represent the realities of marginalized communities. The svyset program allows us to account for the clustered attributes of the sampling strategy. This suggests that individuals within the same cluster (e.g., geographic regions or communities) may demonstrate more similarity to each other than to those in different clusters. By specifying the clustering variable in our study, we adjusted the standard errors to account for intra-cluster correlation, yielding more reliable statistical inferences. The DHS employs a stratified sampling procedure to ensure representation across various demographic and geographic categories. Integrating the stratification variable into our svyset command maintained the sample structure and ensured the generalisability of our findings to the broader population.

## Ethical Statement

Ethical clearance was not sought for this study as it utilized secondary data freely available in the public domain through the DHS program. The DHS data is fully anonymized, with all personally identifiable information removed to protect the privacy of respondents. The original survey adhered to strict ethical protocols, including obtaining informed consent from all participants before data collection. Participants were made aware of the voluntary nature of their participation and were assured of the confidentiality of their responses. Additionally, the DHS program is reviewed and approved by the Institutional Review Board (IRB) of ICF International and, where applicable, by local IRBs in the respective countries where the surveys are conducted. While IPV is a sensitive topic, the DHS ensures that questions related to IPV are administered with care, typically in private settings to minimize distress or harm to respondents. More details about the ethical protocols followed in the DHS surveys can be accessed at (http://goo.gl/ny8T6X).

## Results

The socio-demographic characteristics of married women in Sierra Leone reveal key insights into IPV. Most women are aged 25–39 (57.6%), with a significant proportion (60.7%) having no formal education, highlighting potential barriers to accessing information and resources. Most women (81.4%) are currently working. Yet, access to family planning (FP) information through media is limited, with only 1.5%, 28.4%, and 8.1% hearing about FP in newspapers, on the radio, and TV, respectively. Nearly half of the women (48.1%) experience three or more partner-controlling behaviours, and 36.1% justify three or more reasons for wife beating, indicating entrenched gender norms. While 28.3% report exposure

to interparental violence, 17.9% have partners who drink alcohol, a known IPV risk factor. Socioeconomic disparities are evident, with 43.4% of women falling into the poorest and poorer wealth categories. Additionally, most households are male-headed (76.6%), and the majority of women reside in rural areas (63.0%), with provincial distribution reasonably even across the five provinces (Table 1).

Table 2 shows the prevalence of different forms of IPV among married women of reproductive age in Sierra Leone. Physical violence affects 38.2% of women, while sexual violence is less common, reported by 6.2% of women. Emotional violence is highly prevalent, experienced by 45.9% of women. Overall, 56.0% of married women reported experiencing at least one form of IPV, underscoring the widespread nature of violence within intimate relationships.

The bivariate analysis reveals significant associations between socio-demographic factors and IPV among married women in Sierra Leone. Younger women (ages 20–29) face higher rates of IPV and emotional violence, with prevalence decreasing after age 35, while sexual violence peaks at ages 30–34 (9.2%) and declines sharply by 45–49 (1.5%). Women with primary or secondary education report higher rates of IPV and emotional violence compared to those with no or higher education. IPV is strongly linked to partner-controlling behaviours, with IPV (71.5%), emotional (61.2%), and sexual violence (8.8%) most common among women experiencing three or more controlling behaviours. Justifying wife beating and exposure to interparental violence significantly increase IPV prevalence, as does having a partner who drinks alcohol, with affected women reporting much higher rates of all violence types. Provincially, the Northwestern and Western area show the highest IPV prevalence, notably (69.3% in Northwestern) and emotional violence (58.7% in Northwestern), while sexual violence is most prevalent in the Western area (10.3%). Other factors, such as employment status, wealth, household head's sex, and residence, show limited or no significant associations, emphasizing the need to address partner behaviours, harmful gender norms, and provincial disparities to reduce IPV (Table 3).

## Spatial pattern of Intimate partner violence in Sierra Leone

The spatial autocorrelation analysis of intimate partner violence in Sierra Leone, as presented in Table 4, revealed statistically significant spatial clustering for both physical and emotional violence. In contrast, sexual intimate partner violence did not show statistically significant clustering. The p-value of 0.000000 for physical and emotional intimate partner violence indicates that the observed spatial patterns are unlikely to be due to random chance. Conversely, the p-values of 0.092193 suggest that the spatial pattern of sexual IPV may indeed be attributable to chance.

Moran's I statistic values for all three forms of intimate partner violence demonstrate positive spatial autocorrelation, suggesting that areas with higher incidences of IPV are more likely to be clustered together. Furthermore, the Z-scores of 5.219904 for physical violence (PV) and 10.250871 for emotional violence (EV) underscore the strong clustering effect in these areas. In contrast, the Z-score of 1.683941 for sexual violence (SV) indicates a weaker clustering effect in those regions.

## Hotspot analysis

The hotspots analysis of physical, emotional, and sexual IPV varied considerably across the 16 districts of Sierra Leone. In all three maps, red areas indicate hotspots, while blue ones represent cold spots. Hotspot clusters (i.e., districts with a higher occurrence of physical IPV) of physical IPV were observed primarily in Western area urban, Western area rural, Kambia, and Karene districts. By contrast, cold spots (i.e., clusters of districts with lower occurrence of physical IPV) of physical IPV were primarily concentrated in Bo, Kenema and Bombali districts (see Fig 1, panel A).

However, hotspot clusters (i.e., clusters of districts with a higher occurrence of emotional IPV) of emotional IPV were observed primarily in Western Area urban, Western Area rural, and a few areas of Kambia, Karene and Tonkolili districts. By contrast, cold spots (i.e., clusters of districts with lower occurrence of emotional IPV) of emotional IPV were identified in five districts from Bo, Kenema, Bombali, Kailahun, and some parts of Bonthe districts (see Fig 1, panel B).

**Table 1. Socio-demographic characteristics of intimate partner violence among married women in Sierra Leone (n = 3,611).**

| Variables | Weighted sample | Weighted percentage |
|---|---|---|
| **Women's age (Year)** | | |
| 15-19 | 170 | 4.7 |
| 20-24 | 498 | 13.8 |
| 25-29 | 719 | 19.9 |
| 30-34 | 640 | 17.7 |
| 35-39 | 721 | 20.0 |
| 40-44 | 460 | 12.7 |
| 45-49 | 402 | 11.1 |
| **Educational level** | | |
| No education | 2,193 | 60.7 |
| Primary | 467 | 12.9 |
| Secondary | 827 | 22.9 |
| Higher | 123 | 3.4 |
| **Currently working** | | |
| No | 671 | 18.6 |
| Yes | 2,940 | 81.4 |
| **Heard about FP in a newspaper/magazine last few months** | | |
| No | 3,556 | 98.5 |
| Yes | 54 | 1.5 |
| **Heard FP on the radio last few months** | | |
| No | 2,586 | 71.6 |
| Yes | 1,025 | 28.4 |
| **Heard about family planning on TV in the last few months** | | |
| No | 3,318 | 91.9 |
| Yes | 292 | 8.1 |
| **Partner controlling behaviours** | | |
| None | 751 | 20.8 |
| One-two | 1,122 | 31.1 |
| Three+ | 1,737 | 48.1 |
| **Justification of wife-beating** | | |
| None | 1,706 | 47.3 |
| One-two | 601 | 16.7 |
| Three+ | 1,303 | 36.1 |
| **Exposure to interparental violence** | | |
| No | 2,590 | 71.7 |
| Yes | 1,021 | 28.3 |
| **Husband/partner drinks alcohol** | | |
| no | 2,966 | 82.1 |
| yes | 645 | 17.9 |
| **Wealth index combined** | | |
| Poorest | 800 | 22.2 |
| Poorer | 764 | 21.2 |
| Middle | 739 | 20.5 |
| Richer | 668 | 18.5 |
| Richest | 639 | 17.7 |

*(Continued)*

**Table 1.** (Continued)

| Variables | Weighted sample | Weighted percentage |
|---|---|---|
| **Sex of household head** | | |
| Male | 2,766 | 76.6 |
| Female | 844 | 23.4 |
| **Residence** | | |
| Urban | 1,338 | 37.0 |
| Rural | 2,273 | 63.0 |
| **Province** | | |
| Eastern | 753 | 20.8 |
| Northern | 824 | 22.8 |
| Northwestern | 621 | 17.2 |
| Southern | 718 | 19.9 |
| Western | 694 | 19.2 |

Furthermore, Hotspot clusters (i.e., clusters of districts with a higher occurrence of sexual IPV) of sexual IPV were observed primarily in the Western Area Urban, Western Area Rural, and a few areas of Karene and Falaba districts. Conversely, cold spots (i.e., clusters of districts with lower occurrence of sexual IPV) of sexual IPV were identified in five districts from Bo, Kenema, and Kailahun districts **(see** Fig 1**, panel C).**

### Kriging Interpolation

The Kriging interpolation analysis was carried out to predict higher IPV districts in Sierra Leone. The predicted percentage of women's IPV increases from yellow to red, where districts marked in red areas indicate districts with a high occurrence of IPV, while areas with green indicate a low occurrence of IPV.

However, the Western area urban, Western area rural, Kambia, Karene and Falaba districts had the highest percentage of physical IPV **(see** Fig 2**, panel A)**. Moreover, the Western area urban, Western area rural, Karene and Tonkolili districts had the highest percentage of emotional IPV **(see** Fig 2**, panel B)**. Furthermore, Falaba and some parts of Karene had the highest percentage of sexual IPV **(see** Fig 2**, panel C)**.

The analysis identifies several factors significantly associated with intimate partner violence among married women in Sierra Leone. Younger women, particularly those aged 25–29, face higher odds of IPV compared to women aged 15–19, though the association weakens with age. Partner-controlling behaviours strongly increase IPV risk, with women experiencing three or more controlling behaviours having 9 times higher odds than those with no controlling behaviours. Justifying wife beating and exposure to interparental violence also significantly elevates IPV risk, as does having a partner who consumes alcohol, which more than doubles the odds of IPV. Provincial disparities are evident, with women in the Northwestern and Western area facing significantly higher IPV risks compared to the Eastern province. While some variables, such as partner controlling behavior, exposure to interparental violence, and husband's alcohol consumption, were found to have strong and statistically significant associations with IPV, others, including educational attainment and wealth index, were non-significant in the adjusted models.

### Goodness-of-fit of the models

The goodness-of-fit of the models presented in Table 5 were assessed using several statistical metrics, including the Akaike Information Criterion (AIC), log-likelihood values, and Wald chi-square tests. These metrics provide insight into the relative quality of each model and its ability to explain the observed data.

**1. Akaike Information Criterion (AIC).** The AIC values progressively decreased across the models, indicating improved model fit as additional variables were included. The empty model (Model I) had the highest AIC value [4672.93],

**Table 2. Prevalence of intimate partner violence among married reproductive aged women in Sierra Leone.**

| Variables | Weighted sample | Weighted percentage |
|---|---|---|
| **Physical violence** | | |
| No | 2,230 | 61.8 |
| Yes | 1,380 | 38.2 |
| **Sexual violence** | | |
| No | 3,388 | 93.8 |
| Yes | 223 | 6.2 |
| **Emotional violence** | | |
| No | 1,953 | 54.1 |
| Yes | 1,657 | 45.9 |
| **Intimate partner violence** | | |
| No | 1,589 | 44.0 |
| Yes | 2,022 | 56.0 |

reflecting its limited explanatory power due to the absence of covariates. Model IV, the final model, had the lowest AIC value [3990.38], suggesting that it provided the best fit to the data among the models. A lower AIC value indicates a better balance between model complexity and goodness-of-fit, as it penalizes unnecessary complexity.

**2. Log-Likelihood.** Similarly, the log-likelihood values improved across the models, with Model IV having the highest log-likelihood [−1968.19]. Higher log-likelihood values indicate that the model is better at predicting the observed data. This improvement reflects the inclusion of additional covariates in Model IV, which enhanced its explanatory power.

**3. Wald Chi-Square Tests.** The Wald chi-square test results demonstrate significant improvements in model fit as covariates were added. For instance, Model II had a Wald chi-square value of [317.55***], while Model IV showed the highest value of [364.31***], indicating that the inclusion of additional variables significantly contributed to explaining the variance in IPV.

**4. Random Effects.** ICC values decreased slightly across the models, from 0.24 in Model I to 0.22 in Model IV. This reduction suggests that the inclusion of individual and contextual covariates in the final model accounted for some of the unexplained variance at the cluster level, improving the overall fit.

## Discussion

This study examines the spatial mapping and factors associated with IPV among women of childbearing age (15–49 years) in Sierra Leone in 2019. The study revealed an overall IPV prevalence of 56%, with physical, sexual, and emotional violence accounting for 38%2%, 6.2%, and 45.9%, respectively. Hotspot districts were identified in the Western area (urban and rural areas) and the Northwestern province (Kambia and Karene). At the same time, Bo, Kenema, and Bombali, the provincial headquarters of the Northern, Eastern, and Southern provinces, were found as cold spot districts. Factors associated with IPV included women aged 25–29, those with one-two partners exhibiting controlling behaviour, and those who provided one-two justifications for wife beating. Furthermore, women exposed to interparental violence and those who resided in the Northwestern, Northern, and Western area had a higher likelihood of experiencing IPV.

The overall IPV prevalence found in this study was higher than the average reported in a multicountry study involving 26 sub-Saharan African countries, including Sierra Leone, which recorded a prevalence of 42.6%. The prevalence of emotional and physical violence in that study (30.2% and 30.6%, respectively) was lower than the rates observed in this study, whereas the prevalence of sexual violence (12.6%) was higher than what was found here [23]. Additionally, the multicountry study reported an IPV prevalence of 59.9% in Sierra Leone, which was similar to the findings of this study [23]. Another study on the prevalence, correlates, and trends of IPV against women in Sierra Leone reported a lifetime IPV prevalence

**Table 3. Bivariate analysis of IPV among married reproductive aged women in Sierra Leone.**

| Variables | Physical violence | Emotional violence | Sexual violence | IPV |
|---|---|---|---|---|
| | % (95% CI) | % (95% CI) | %(95% CI) | %(95% CI) |
| **Women's age (Year)** | <0.001 | <0.001 | P=0.002 | <0.001 |
| 15-19 | 6.3 [4.7,8.4] | 39.7 [31.8,48.3] | 8.5 [4.7,14.9] | 56.2 [47.5,64.5] |
| 20-24 | 26.2 [22.8,29.8] | 51.6 [46.2,57.0] | 7.0 [4.8,10.1] | 63.6 [58.7,68.6] |
| 25-29 | 38.7 [33.5,44.2] | 52.7 [47.7,57.7] | 7.1 [5.2,9.7] | 65.2 [60.1,69.9] |
| 30-34 | 36.1 [31.6,40.8] | 49.0 [44.2,53.8] | 9.2 [6.1,13.6] | 58.3 [53.3,63.1] |
| 35-39 | 33.6 [29.4,38.1] | 43.2 [38.6,47.9] | 4.2 [3.0,6.2] | 52.4 [47.8,56.8] |
| 40-44 | 30.2 [24.5,36.6] | 39.2 [33.1,45.7] | 5.8 [3.7,9.0] | 46.9 [40.5,53.3] |
| 45-49 | 19.9 [15.5,25.3] | 36.7 [31.3,42.6] | 1.5 [0.5,4.1] | 43.3 [37.8,49.0] |
| **Educational level** | <0.001 | P=0.002 | P=0.150 | P=0.001 |
| No education | 32.1 [29.5,34.8] | 42.8 [39.9,45.8] | 5.7 [4.5,7.3] | 52.7 [49.9,55.5] |
| Primary | 27.8 [23.5,32.5] | 54.1 [48.4,59.6] | 6.2 [4.1,9.3] | 63.2 [58.4,67.8] |
| Secondary | 19.1 [16.5,22.0] | 51.0 [45.8,56.2] | 7.9 [5.5,11.2] | 61.9 [56.5,67.1] |
| Higher | 22.2 [15.8,30.3] | 35.8 [25.5,47.5] | 2.5 [0.9,7.1] | 47.1 [35.7,58.8] |
| **Currently working** | <0.001 | P=0.763 | P=0.033 | P=0.774 |
| No | 16.7 [14.1,19.7] | 46.6 [41.3,51.9] | 8.5 [5.9,12.3] | 56.7 [51.4,61.7] |
| Yes | 30.5 [28.0,33.2] | 45.7 [43.1,48.4] | 5.6 [4.6,6.9] | 55.9 [53.1,58.6] |
| **Heard about FP in a newspaper/magazine last few months** | P=0.243 | P=0.432 | P=0.432 | P=0.432 |
| No | 26.2 [24.2,28.3] | 55.9 [53.3,58.5] | 55.9 [53.3,58.5] | 55.9 [53.3,58.5] |
| Yes | 32.7 [22.2,45.2] | 62.7 [45.3,77.4] | 62.7 [45.3,77.4] | 62.7 [45.3,77.4] |
| **Heard FP on the radio last few months** | P=0.005 | <0.001 | P=0.140 | <0.001 |
| No | 24.6 [22.7,26.6] | 42.1 [39.5,44.8] | 5.6 [4.4,6.9] | 52.5 [49.8,55.2] |
| Yes | 30.5 [26.5,34.8] | 55.5 [51.0,59.9] | 7.8 [5.2,11.4] | 64.9 [60.4,69.1] |
| **Heard about family planning on TV in the last few months** | P=0.566 | P=0.523 | P=0.351 | P=0.214 |
| No | 26.1 [24.1,28.3] | 45.7 [43.2,48.2] | 6.0 [4.9,7.5] | 55.6 [53.0,58.1] |
| Yes | 27.8 [22.4,34.0] | 48.6 [39.8,57.4] | 7.9 [4.6,13.2] | 60.6 [52.5,68.1] |
| **Partner controlling behaviors** | <0.001 | <0.001 | P=0.001 | <0.001 |
| None | 5.3 [3.9,7.3] | 17.8 [13.9,22.5] | 1.3 [0.4,4.1] | 25.8 [21.2,31.0] |
| One-two | 34.2 [29.9,38.9] | 41.1 [37.3,45.0] | 5.3 [3.7,7.6] | 52.3 [48.1,56.4] |
| Three+ | 50.0 [46.8,53.2] | 61.2 [57.8,64.4] | 8.8 [7.3,10.7] | 71.5 [68.4,74.4] |
| **Justification of wife-beating** | <0.001 | <0.001 | P=0.037 | <0.001 |
| None | 20.6 [17.8,23.7] | 40.2 [36.7,43.7] | 5.2 [3.8,7.1] | 49.7 [46.1,53.4] |
| One-two | 32.6 [29.0,36.4] | 50.4 [45.1,55.8] | 5.5 [3.8,7.9] | 62.9 [57.8,67.6] |
| Three+ | 32.4 [29.2,35.7] | 51.3 [47.6,55.0] | 7.8 [6.1,9.8] | 61.1 [57.4,64.7] |
| **Exposure to interparental violence** | <0.001 | <0.001 | P=0.357 | <0.001 |
| No | 22.4 [20.3,24.6] | 40.5 [37.6,43.4] | 5.9 [4.6,7.5] | 50.2 [47.2,53.1] |
| Yes | 35.4 [31.6,39.5] | 59.7 [54.7,64.5] | 6.9 [5.2,9.0] | 70.8 [66.4,74.8] |
| **Husband/partner drinks alcohol** | <0.001 | <0.001 | P=0.004 | <0.001 |
| No | 34.0 [31.5,36.5] | 41.3 [38.6,43.9] | 5.4 [4.2,6.8] | 51.6 [48.8,54.3] |
| Yes | 57.9 [52.1,63.4] | 67.2 [62.3,71.8] | 9.9 [7.5,13.0] | 76.4 [72.0,80.4] |
| **Wealth index combined** | P=0.006 | P=0.115 | P=0.102 | P=0.355 |
| Poorest | 30.8 [26.3,35.7] | 40.1 [36.2,44.0] | 3.9 [2.7,5.5] | 51.4 [46.9,55.8] |
| Poorer | 30.0 [26.3,34.1] | 47.6 [43.1,52.2] | 6.1 [4.2,8.8] | 57.8 [52.9,62.5] |
| Middle | 27.3 [23.9,31.0] | 48.3 [43.9,52.7] | 6.0 [4.3,8.3] | 57.0 [52.4,61.5] |
| Richer | 22.5 [19.2,26.1] | 47.6 [42.5,52.8] | 7.2 [4.8,10.8] | 57.7 [52.3,62.9] |
| Richest | 22.5 [18.1,27.5] | 46.6 [39.3,54.0] | 8.2 [5.4,12.4] | 56.8 [49.3,63.9] |

*(Continued)*

**Table 3.** (Continued)

| Variables | Physical violence | Emotional violence | Sexual violence | IPV |
|---|---|---|---|---|
| | % (95% CI) | % (95% CI) | %(95% CI) | %(95% CI) |
| **Sex of household head** | <0.001 | P=0.856 | P=0.436 | P=0.246 |
| Male | 29.9 [27.6,32.3] | 46.0 [43.3,48.7] | 5.9 [4.8,7.3] | 56.7 [53.9,59.4] |
| Female | 18.0 [15.3,21.2] | 45.6 [41.0,50.2] | 6.9 [4.7,10.1] | 53.8 [49.1,58.4] |
| **Residence** | P=0.002 | P=0.213 | P=0.219 | P=0.455 |
| Urban | 22.6 [19.7,25.7] | 48.1 [43.3,53.0] | 7.2 [5.1,10.2] | 57.4 [52.3,62.4] |
| Rural | 29.4 [26.7,32.2] | 44.6 [41.8,47.4] | 5.6 [4.4,7.0] | 55.2 [52.4,58.0] |
| **Province** | <0.001 | <0.001 | <0.001 | <0.001 |
| Eastern | 19.1 [16.4,22.2] | 38.0 [33.6,42.7] | 2.8 [1.8,4.3] | 46.9 [42.1,51.8] |
| Northern | 31.3 [26.1,37.0] | 42.8 [37.6,48.1] | 5.9 [3.8,8.9] | 56.6 [51.4,61.6] |
| Northwestern | 35.5 [30.6,40.7] | 58.7 [53.1,64.1] | 9.9 [7.1,13.6] | 69.3 [64.1,74.0] |
| Southern | 20.5 [17.7,23.7] | 38.6 [34.9,42.5] | 2.8 [1.9,4.3] | 47.4 [43.4,51.4] |
| Western | 26.7 [22.1,31.9] | 54.1 [46.3,61.7] | 10.3 [6.6,15.7] | 62.2 [53.7,70.0] |

**Table 4. Spatial autocorrelation of Intimate partner violence in Sierra Leone, 2019.**

| Intimate partner violence | Moran's Index | Variance | Z-score | p-value |
|---|---|---|---|---|
| Physical | 0.164548 | −0.001016 | 5.219904 | 0.000000 |
| Emotional | 0.325073 | −0.001017 | 10.250871 | 0.000000 |
| Sexual | 0.051621 | −0.001006 | 1.683941 | 0.092193 |

of 60.8%, slightly exceeding the prevalence found in this study [30]. More concerningly, the IPV prevalence observed in this study surpasses the global average of 27% and the sub-Saharan Africa prevalence of 33% reported by the World Health Organization [5]. Several factors may explain the high prevalence of IPV in Sierra Leone. These include a weak judiciary system, limited prioritisation of IPV by stakeholders, low socioeconomic status, cultural norms, lower educational levels, widespread acceptance of IPV as normal within the population, history of civil war, and health emergencies such as Ebola and COVID-19 [30,46–48]. The interaction of these factors likely contributes to the higher IPV prevalence observed in this study. Considering that IPV significantly contributes to the disease burden and imposes high economic costs [5,49], this situation is particularly troubling for Sierra Leone, where over half of the population lives below a national poverty line of $1.25 per day, and the healthcare system is already under significant strain to address emerging public health threats.

This study identified the Western area urban and rural districts in the Western area and the Kambia and Karene districts in the Northwestern province as hotspot areas for IPV in Sierra Leone. While the Western Area Urban and Rural districts serve as major administrative centres for most law enforcement and legal institutions, they are densely populated and struggle with inadequate housing infrastructure to support the growing population. These factors may have contributed to the higher prevalence of IPV in these areas. Conversely, although Kambia and Karene, both rural districts, are sparsely populated, they face significant challenges such as high illiteracy rates, entrenched cultural norms, limited employment opportunities, weaker law enforcement systems, limited access to healthcare and IPV support services, and low awareness of IPV and women's rights [50]. These factors may have contributed to the higher IPV prevalence observed in these districts. Addressing these challenges is critical in reducing IPV in these districts. Additionally, the study identified Bo, Kenema, and Bombali districts as cold spot areas for IPV. These districts serve as the regional headquarters towns for the Southern, Eastern, and Northern regions of Sierra Leone. Unlike the Western area urban, these districts have relatively smaller populations [51] and adequate access to resources that help mitigate IPV. The effective use of

## Hot spot and cold spot analysis of IPV among married women in Sierra Leone *2019.*

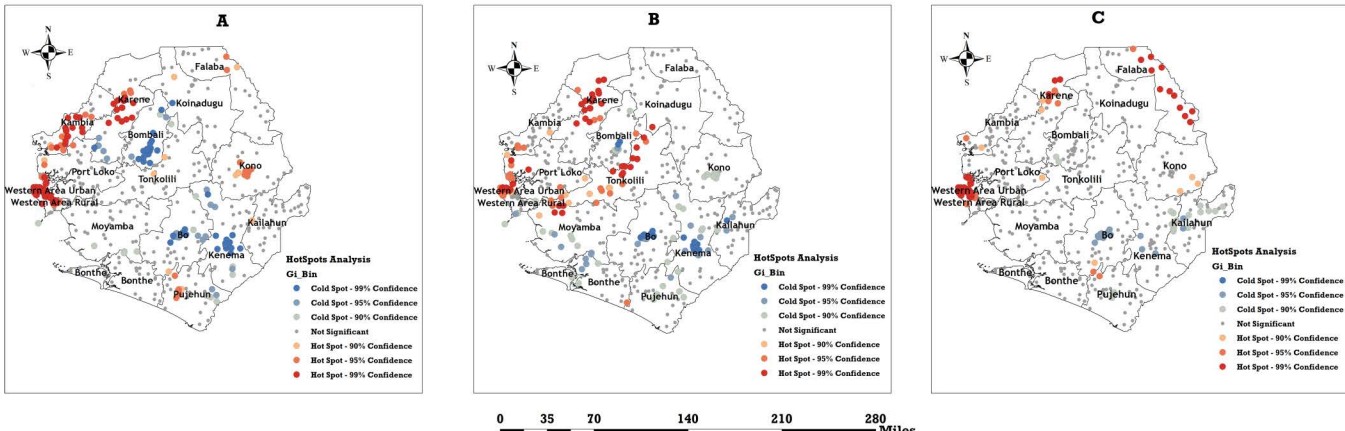

**Fig 1. Hotspots and cold spot analysis of intimate partner violence among women in Sierra Leone, 2019.** The red areas indicate hotspots (districts with a higher occurrence of IPV), while the blue areas represent cold spots (districts with a lower occurrence of IPV). Panel A: Physical Violence. Panel B: Emotional violence. Panel C: Sexual violence. Spatial Data Respository, The Demographic and Health Surveys Program. ICF International. Funded by the united States Agency for International Development (USAID). Available from spatialdata.dhsprogram. [Accessed 04 December 2024]. Created by the authors and published under the CC BY 4.0 license.

## Spatial interpolation of intimate partner violence among married women in Sierra Leone *2019*

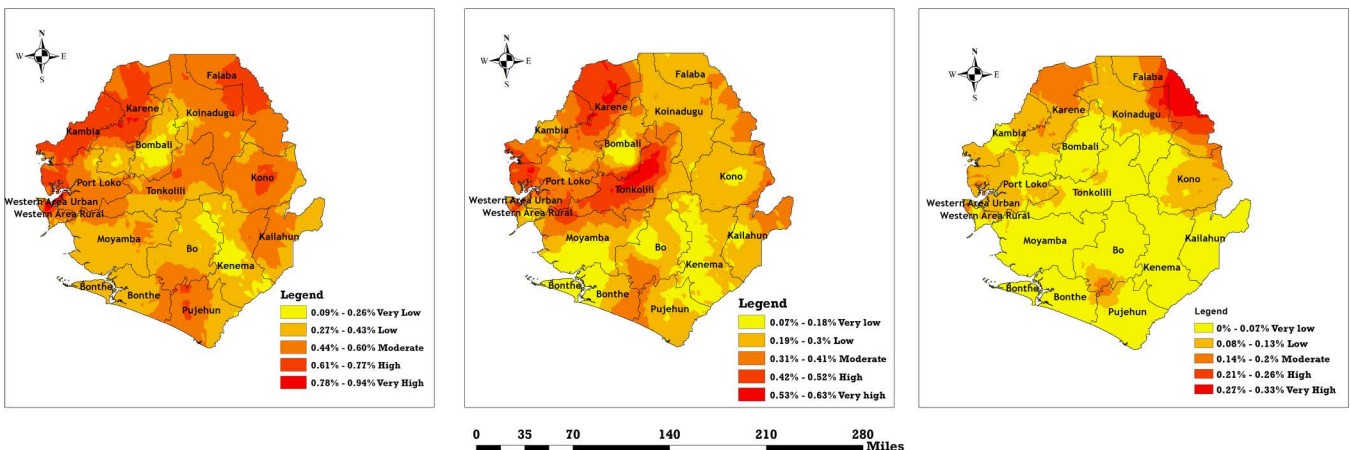

**Fig 2. Kriging interpolation to predict IPV among women in Sierra Leone, 2019.** The red indicates districts with higher occurrence of IPV, and the yellow indicates districts with lower occurrence of IPV. Spatial Data Respository, The Demographic and Health Surveys Program. ICF International. Funded by the united States Agency for International Development (USAID). Available from spatialdata.dhsprogram. [Accessed 04 December 2024]. Created by the authors and published under the CC BY 4.0 license.

these resources may have contributed to the lower IPV rates observed in these districts. Hence, strategies implemented in these districts will be crucial to expand to hotspot districts to reduce the burden of IPV in Sierra Leone.

Results further show that married women aged 25–29 years had 1.8 times higher odds of experiencing IPV compared to those 15–19 years, and the association was statistically significant. This finding contrast a multicountry analysis on the

**Table 5. Factors associated with IPV among married women in Sierra Leone.**

| Variables | Model I Empty model | Model II aOR [95% CI] | Model III aOR [95% CI] | Model IV aOR [95% CI] |
|---|---|---|---|---|
| **Fixed effect results** | | | | |
| **Women's age (years)** | | | | |
| 15-19 | | 1.00 | | 1.00 |
| 20-24 | | 1.68* [1.03,2.76] | | 1.60 [0.97,2.62] |
| 25-29 | | 1.90* [1.15,3.16] | | 1.76* [1.06,2.93] |
| 30-34 | | 1.49 [0.89,2.48] | | 1.36 [0.81,2.28] |
| 35-39 | | 1.26 [0.75,2.12] | | 1.14 [0.68,1.92] |
| 40-44 | | 1.03 [0.59,1.81] | | 0.90 [0.51,1.58] |
| 45-49 | | 0.98 [0.56,1.72] | | 0.86 [0.49,1.52] |
| **Educational attainment** | | | | |
| No education | | 1.00 | | 1.00 |
| Primary | | 1.14 [0.85,1.65] | | 1.18 [0.89,1.56] |
| Secondary | | 1.17 [0.89,1.54] | | 1.12 [0.84,1.50] |
| Higher | | 0.72 [0.42,1.25] | | 0.67 [0.38,1.17] |
| **Currently working** | | | | |
| No | | 1.00 | | 1.00 |
| Yes | | 1.14 [0.86,1.51] | | 1.16 [0.88,1.54] |
| **Partner controlling behaviour** | | | | |
| None | | 1.00 | | 1.00 |
| One-tw0 | | 3.75 ** * [2.79,5.05] | | 3.57 *** [2.65,4.82] |
| Three+ | | 9.78*** [7.00,13.64] | | 9.09*** [6.54,12.63] |
| **Justification of wife-beating** | | | | |
| None | | 1.00 | | 1.00 |
| One-two | | 1.46* [1.05,2.04] | | 1.43* [1.01,2.04] |
| Three+ | | 1.26 [1.00,1.60] | | 1.16 [0.91,1.46] |
| **Exposure to interparental violence** | | | | |
| N0 | | 1.00 | | 1.00 |
| Yes | | 3.00*** [2.31,3.90] | | 2.74*** [2.13,3.54] |
| **Husband/partner alcohol consumption** | | | | |
| No | | | 1.00 | 1.00 |
| Yes | | | 3.38*** [2.56,4.46] | 2.10*** [2.18,4.12] |
| **Wealth index** | | | | |
| Poorest | | | 1.00 | |
| Poorer | | | 1.14 [0.83,1.56] | 1.04 [0.73,1.47] |
| Middle | | | 1.05 [0.78,1.41] | 1.05 [0.77,1.44] |
| Richer | | | 1.17 [0.79,1.73] | 1.11 [0.74,1.67] |
| Richest | | | 0.86 [0.55,1.33] | 0.80 [0.49,1.32] |
| **Residence** | | | | |
| Urban | | | 1.00 | 1.00 |
| Rural | | | 1.04 [0.73,1.49] | 0.98 [0.67,1.42] |
| **Region** | | | | |
| Eastern | | | 1.00 | 1.00 |
| Northern | | | 1.31 [0.94,1.82] | 1.49* [1.05,2.11] |
| Northwestern | | | 2.58*** [1.84,3.61] | 2.11*** [1.46,3.08] |
| Southern | | | 0.93 [0.69,1.24] | 0.82 [0.59,1.16] |

*(Continued)*

**Table 5.** (Continued)

| Variables | Model I Empty model | Model II aOR [95% CI] | Model III aOR [95% CI] | Model IV aOR [95% CI] |
|---|---|---|---|---|
| Western | | | 2.16** [1.34,3.48] | 2.33*** [1.41,3.86] |
| **Random effect model** | | | | |
| PSU variance (95% CI) | 1.06 [0.81, 1.39] | 1.12 [0.85, 1.48] | 0.87 [0.64, 1.17] | 0.94 [0.68, 1.29] |
| ICC | 0.24 [0.20, 0.30] | 0.25 [0.21, 0.31] | 0.21 [0.16, 0.26] | 0.22 [0.17, 0.28] |
| Wald chi-square | Reference | 317.55*** | 133.20*** | 364.31*** |
| **Model fitness** | | | | |
| Log-likelihood | −2334.46 | −2028.15 | −2250.52 | −1968.19 |
| AIC | 4672.93 | 4090.31 | 4525.05 | 3990.38 |
| N | 4,055 | 4,055 | 4,055 | 4,055 |
| Number of clusters | 574 | 574 | 574 | 574 |

aOR= adjusted odds ratios; CI=Confidence Interval;

*p<; 0.05,

**p<; 0.01,

***p<; 0.001; 1.00=Reference category; PSU=Primary Sampling Unit; ICC=Intra-Class Correlation; AIC=Akaike's Information Criterion

magnitude and determinants of intimate partner violence against women conducted among 11 East African countries, where the authors reported women 25–29 years had no association with intimate partner violence [22]. In Sierra Leone, the increased odds of IPV among women aged 25–29 can be attributed to the considerable economic challenges they encounter, which may impact their choices and behaviours. For example, engaging in multiple relationships and stronger commitments to marriage or cohabitation may amplify their risk of experiencing IPV. Tackling these underlying issues could contribute to reducing IPV in Sierra Leone.

This study also revealed that married women with more than one partner exhibiting controlling behaviour had higher odds of experiencing IPV compared to those with none. This finding aligns with a study conducted in Uganda, where researchers reported that women with partners who displayed controlling behaviour were 5.2 times more likely to experience IPV, with the association being statistically significant [52]. Similarly, a multi-country study by the World Health Organization reported comparable results [53]. In Uganda [52], researchers proposed that these findings could be attributed to cultural traditions in which men are usually required to provide a marriage fee, known as a "dowry." This practice confers significant authority on men and bolsters male dominance. Since Uganda and Sierra Leone exhibit comparable cultural and traditional practices, this reasoning may help explain the results observed in this study.

Consistent with a study conducted in Uganda [54], this study also found that women with more than one justification of wife beating had higher odds of experiencing IPV compared with those with none. In Uganda [54], the authors discussed that the reported findings for wife-beating could be attributed to the deep-rooted cultural norms and belief of male dominance, the perception of certain women to be beating as a form of true love from their spouses, and exposure to violence as possible explanations for their findings. In Sierra Leone, the stated reasons could explain the observed findings found in this study. Other possible reasons could be attributed to the prolonged eleven-year civil war, the Ebola outbreak, and the COVID-19 pandemic potentially exacerbating IPV.

Similar to a multi-level country analysis conducted for East African countries [55], this study found that women whose husbands consumed alcohol were 2 times more likely to experience IPV than their counterparts whose husbands did not. In the East African study, the authors discussed that although a statistically significant finding was found in their study and several other studies, establishing a causal relationship will be difficult because individuals may provide misinformation. Additionally, the authors discussed that women of husbands who drink alcohol are vulnerable to IPV because

whenever their husband drinks alcohol, it affects their reasoning ability to recognize fault and violation against women's rights. With 14.3% of men and 5.2% of women population engaging in episodic drinking in Sierra Leone [56], the finding is worrisome, implying that the rate of violence remains high if urgent measures are not taken to address this issue. Hence, stakeholders are recommended to develop policies on alcohol abusers, and fines should be levied on defaulters, conduct mass sensitization using different fora on the link between alcohol and IPV, and intensify treatment services for alcohol abusers.

Lastly, women who experienced interparental violence were 2.7 times more likely to experience IPV than those who did not. This finding was consistent with a multilevel country-level analysis conducted for 27 Sub-Saharan African countries where the authors reported that women who reported their mothers experiencing IPV were 2.7 times more likely to face IPV [57]. Since interparental violence is reported as a risk factor for IPV [5], this finding is worrisome for Sierra Leone, where there are weaker structures for addressing violence against women. It is, therefore, imperative for stakeholders to strengthen key structures, including mental health support services, intensify awareness-raising activities on the negative impacts of IPV, and encourage victims to report cases. Families should encourage regular communication on the dangers of IPV and ways of abstaining from it.

## Policy and Practice Implications

The findings of this study underscore the need for targeted interventions to address IPV in Sierra Leone, where prevalence remains alarmingly high. Geographic mapping of IPV prevalence provides a valuable opportunity for local authorities and NGOs to prioritize resources and implement interventions in hotspot districts such as the Western and Northwestern provinces. This tool can guide the allocation of funding, deployment of community-based education campaigns, and establishment of IPV support centers in high-risk areas. Meanwhile, cold spot districts, where IPV prevalence is comparatively lower, can serve as models for best practices in prevention and response strategies.

Strengthening the enforcement of existing IPV laws is critical to addressing the issue. While Sierra Leone has legal frameworks to combat violence against women, enforcement remains weak due to limited resources and societal acceptance of IPV. Policymakers should focus on building the capacity of law enforcement agencies through training on IPV sensitivity and the establishment of specialized units to handle IPV cases. Complementary public awareness campaigns can educate communities on the legal consequences of IPV, encourage survivors to seek justice, and reduce stigma.

Challenging cultural norms that justify wife-beating and controlling behaviors among partners is essential for long-term change. Community-based initiatives, such as dialogues with traditional leaders, religious figures, and local influencers, can help shift societal attitudes. Engaging men and boys through gender-transformative education programs can further reduce controlling behaviors and promote healthier relationships. These efforts should be complemented by programs aimed at breaking the cycle of intergenerational violence. Women exposed to interparental violence are at higher risk of experiencing IPV themselves, so trauma-informed care, counseling services, and parenting education are necessary to address this issue. Schools and community centers can also serve as safe spaces for children and adolescents to learn conflict resolution skills and reduce the likelihood of perpetuating IPV as adults.

Province-specific strategies are vital to addressing the disparities in IPV prevalence across Sierra Leone. In hotspot districts, interventions should include establishing IPV response centers equipped with legal, psychological, and medical support services. Mobile clinics can be deployed to provide services in remote areas. In contrast, cold spot districts can document and share their successful IPV prevention strategies for replication in other regions. Multi-sectoral collaboration is equally important, as IPV prevention must be integrated into broader health, education, and social programs. For example, maternal and child health services can screen for IPV and provide referrals to support services. A coordinated response involving government ministries, NGOs, and international organizations is essential for ensuring sustainability and impact.

**Feasibility of policy implementation in Sierra Leone's socio-political context**

Implementing these recommendations requires careful consideration of Sierra Leone's socio-political realities, including limited resources, weak institutional capacity, and deeply entrenched cultural norms. To enhance feasibility, interventions should leverage existing community structures, such as the network of community health workers and local leaders, who can be trained to deliver IPV education and support services. Utilizing these structures reduces costs and ensures that interventions are culturally appropriate and accessible.

Strengthening political will is also essential. Advocacy efforts should focus on engaging policymakers and government officials to prioritize IPV prevention as a public health and human rights issue. Demonstrating the economic and social costs of IPV, such as its impact on productivity and healthcare expenses, can help galvanize political action. Securing sustainable funding through partnerships with international donors and development partners will be critical for bridging gaps in government capacity. NGOs can also play a key role in providing technical assistance and resources for IPV prevention programs.

Finally, community ownership of interventions must be prioritized to ensure success and sustainability. Programs designed with input from local communities are more likely to succeed in Sierra Leone's decentralized governance system. Community-driven initiatives that address IPV will resonate more strongly with the population and foster long-term change.

**Strength and limitations**

The SLDHS provides a robust dataset for examining the spatial mapping and determinants of IPV among married women in Sierra Leone, offering nationally representative data with standardised methodologies that ensure comparability across regions. Its inclusion of detailed IPV-related questions and geospatial data allows for an in-depth analysis of regional variations and associated risk factors. However, the cross-sectional nature of the survey limits the ability to establish causal relationships, and self-reported IPV data may be subject to underreporting due to stigma or fear of disclosure. Additionally, the absence of qualitative insights restricts a deeper understanding of cultural and contextual factors influencing IPV, and the focus on married women excludes other vulnerable groups who may also experience IPV. The study focuses exclusively on married women aged 15–49 years, which may introduce potential biases by excluding unmarried women who may also experience IPV. Unmarried women, including those in cohabiting relationships or dating partnerships, are also at risk of IPV, and their exclusion may limit the generalizability of the findings to all women of reproductive age. This exclusion could result in an underestimation of the overall prevalence of IPV, as unmarried women may face different dynamics of abuse, including those related to less formalized or shorter-term relationships. Additionally, cultural and societal factors influencing IPV may differ between married and unmarried women, potentially leading to variations in risk factors and experiences of IPV. Future studies should aim to include unmarried women to provide a more comprehensive understanding of IPV across diverse relationship contexts.

**Conclusion**

This study underscore the high prevalence of intimate partner violence among married women of childbearing age in Sierra Leone, with significant provincial disparities and associated risk factors. Geographic mapping of IPV prevalence provides a valuable tool for targeted interventions, enabling policymakers, local authorities, and NGOs to focus resources on hotspot districts such as the Western and Northwestern province. Concrete recommendations include strengthening enforcement of IPV laws, challenging cultural norms that justify violence, providing trauma-informed care to women exposed to interparental violence, and integrating IPV prevention into health and social services. To ensure feasibility, interventions must leverage existing community structures, build political will, and secure sustainable funding through partnerships with international donors and NGOs. Community ownership of programs will be critical to addressing IPV in Sierra Leone's socio-political context. Multi-sectoral collaboration and province-specific strategies are essential to reducing IPV and improving the safety and well-being of women across the country.

## Author contributions

**Conceptualization:** Augustus Osborne, Camilla Bangura.

**Data curation:** Augustus Osborne, Lovel Fornah.

**Formal analysis:** Augustus Osborne, Lovel Fornah.

**Methodology:** Augustus Osborne, Lovel Fornah.

**Supervision:** Augustus Osborne.

**Writing – original draft:** Augustus Osborne, Umaru Sesay, Camilla Bangura, Lovel Fornah.

**Writing – review & editing:** Augustus Osborne, Umaru Sesay, Camilla Bangura, Lovel Fornah.

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
