## [Decision Letter · Decision Letter 0]

12 Mar 2025

PONE-D-25-03253Spatial Mapping And Determinants of Intimate Partner Violence Among Married Women In Sierra LeonePLOS ONE

Dear Dr. Osborne,

Thank you for submitting your manuscript to PLOS ONE. After careful consideration, we feel that it has merit but does not fully meet PLOS ONE’s publication criteria as it currently stands. Therefore, we invite you to submit a revised version of the manuscript that addresses the points raised during the review process.

We look forward to receiving your revised manuscript.

Kind regards,

Menen Tsegaw Biadiglgn, MPH

Academic Editor

PLOS ONE

Journal Requirements:

2. We note that Figure 1 and 2 in your submission contain [map/satellite] images which may be copyrighted. All PLOS content is published under the Creative Commons Attribution License (CC BY 4.0), which means that the manuscript, images, and Supporting Information files will be freely available online, and any third party is permitted to access, download, copy, distribute, and use these materials in any way, even commercially, with proper attribution. For these reasons, we cannot publish previously copyrighted maps or satellite images created using proprietary data, such as Google software (Google Maps, Street View, and Earth). For more information, see our copyright guidelines: http://journals.plos.org/plosone/s/licenses-and-copyright.

a. You may seek permission from the original copyright holder of Figure 1 and 2 to publish the content specifically under the CC BY 4.0 license.  

Reviewers' comments:

Reviewer's Responses to Questions

**Comments to the Author**

1. Is the manuscript technically sound, and do the data support the conclusions?

Reviewer #1: Yes

Reviewer #2: Partly

2. Has the statistical analysis been performed appropriately and rigorously? 

Reviewer #1: No

Reviewer #2: Yes

3. Have the authors made all data underlying the findings in their manuscript fully available?

Reviewer #1: Yes

Reviewer #2: Yes

4. Is the manuscript presented in an intelligible fashion and written in standard English?

Reviewer #1: Yes

Reviewer #2: Yes

5. Review Comments to the Author

Reviewer #1: ### 1. Measuring Outcome Variables

How do you measure your outcome variable? Have you employed the same measurement for both spatial and multilevel analyses?

### 2. Choice of Spatial Interpolation Techniques

There are numerous spatial interpolation techniques, such as Inverse Distance Weighting (IDW), Kriging, splines, and natural neighbors. Why did you choose Bayesian spatial Kriging? What are the implications and added value of using this model in the context of your findings?

### 3. Geographic Boundaries and Units of Analysis

What geographic boundaries or unit levels were used in each analysis? Based on the DHS data, these could be at the level of countries, regions (comprising multiple countries) within Sub-Saharan Africa (SSA), census clusters, or GPS points of households. It is crucial to clarify this for accurate interpretation of the spatial analysis.

Your findings indicated a district-level analysis. How did you define districts? Specifically, how did you identify the number of districts, and you mentioned 18 districts—how did you determine this? How many districts were included in the study?

You utilized DHS data, which is derived from a spatial repository that includes enumeration areas for specific countries. How did this data translate into district-level analysis? Please provide a clear description of the level of analysis used.

### 4. Managing Missing Data

How do you manage missing data, including missing attributes and missing coordinate data?

### 5. Study Variables

Please elaborate on your outcome variable. Have you categorized it into binary responses (yes or no)? Clearly indicate this.

### 6. Spatial Autocorrelation

Which spatial matrix did you use to report spatial autocorrelation (e.g., Moran's I)? Please articulate the implications based on your findings.

### 7. Multi-Level Analysis

You conducted Intraclass Correlation Coefficient (ICC) analysis. Given that your analysis is multi-level, it would be beneficial to include Measures of Relative Risk (MOR) and Proportional Change in Variance (PCV). What were your community-level and individual-level variables?

You mentioned explanatory variables; please elaborate further. When performing a multi-level analysis, it is essential to differentiate between these variables.

Reviewer #2: I appreciate the opportunity to review this significant study, which explores an important public health and human rights issue by analyzing the spatial distribution and determinants of intimate partner violence (IPV) among married women in Sierra Leone. Please find my detailed comments attached.

6. PLOS authors have the option to publish the peer review history of their article (what does this mean? ). If published, this will include your full peer review and any attached files.

**Do you want your identity to be public for this peer review?** For information about this choice, including consent withdrawal, please see our Privacy Policy .

Reviewer #1: No

Reviewer #2: **Yes: ** Muhammad Aasim

---

## [Author Response · Author response to Decision Letter 0]

11 Apr 2025

The Editor

PLOS ONE

11 th April 2025

Ref: PONE-D-24-25104

Title: Spatial Mapping And Determinants of Intimate Partner Violence Among

Married Women In Sierra Leone.

Response to Reviewers' comments

Dear Sir/Madam,

We want to express our sincere thanks for painstakingly reviewing our manuscript

and providing valuable comments and suggestions. Please see our point-by-point

response to the reviewers' comments and suggestions. Revisions are highlighted

with track changes in the revised manuscript.

Journal Requirements:

When submitting your revision, we need you to address these additional

requirements.

1. Please ensure that your manuscript meets PLOS ONE's style requirements,

including those for file naming. The PLOS ONE style templates can be found at

https://journals.plos.org/plosone/s/file?id=wjVg/PLOSOne_formatting_sample_main_

body.pdf and

https://journals.plos.org/plosone/s/file?id=ba62/PLOSOne_formatting_sample_title_a

uthors_affiliations.pdf

Response: Thank you. We have ensured that our manuscript meets PLOS ONE style requirements.

2. We note that Figure 1 and 2 in your submission contain [map/satellite] images

which may be copyrighted. All PLOS content is published under the Creative

Commons Attribution License (CC BY 4.0), which means that the manuscript,

images, and Supporting Information files will be freely available online, and any third

party is permitted to access, download, copy, distribute, and use these materials in

any way, even commercially, with proper attribution. For these reasons, we cannot

publish previously copyrighted maps or satellite images created using proprietary

data, such as Google software (Google Maps, Street View, and Earth). For more

information, see our copyright guidelines: http://journals.plos.org/plosone/s/licenses-

and-copyright.

We require you to either (1) present written permission from the copyright holder to

publish these figures specifically under the CC BY 4.0 license, or (2) remove the

figures from your submission:

a. You may seek permission from the original copyright holder of Figure 1 and 2 to

publish the content specifically under the CC BY 4.0 license.

We recommend that you contact the original copyright holder with the Content

Permission Form (http://journals.plos.org/plosone/s/file?id=7c09/content-permission-

form.pdf) and the following text:

“I request permission for the open-access journal PLOS ONE to publish XXX under

the Creative Commons Attribution License (CCAL) CC BY 4.0

(http://creativecommons.org/licenses/by/4.0/). Please be aware that this license

allows unrestricted use and distribution, even commercially, by third parties. Please

reply and provide explicit written permission to publish XXX under a CC BY license

and complete the attached form.”

Please upload the completed Content Permission Form or other proof of granted

permissions as an ""Other"" file with your submission.

In the figure caption of the copyrighted figure, please include the following text:

“Reprinted from [ref] under a CC BY license, with permission from [name of

publisher], original copyright [original copyright year].”

b. If you are unable to obtain permission from the original copyright holder to publish

these figures under the CC BY 4.0 license or if the copyright holder’s requirements

are incompatible with the CC BY 4.0 license, please either i) remove the figure or ii)

supply a replacement figure that complies with the CC BY 4.0 license. Please check

copyright information on all replacement figures and update the figure caption with

source information. If applicable, please specify in the figure caption text when a

figure is similar but not identical to the original image and is therefore for illustrative

purposes only.

The Gateway to Astronaut Photography of Earth (public

domain): http://eol.jsc.nasa.gov/sseop/clickmap/

Maps at the CIA (public domain): https://www.cia.gov/library/publications/the-world-

factbook/index.html and https://www.cia.gov/library/publications/cia-maps-

publications/index.html

USGS EROS (Earth Resources Observatory and Science (EROS) Center) (public

domain): http://eros.usgs.gov/#

Response: Thank you for your input regarding Figures 1 and 2 in our manuscript. We confirm that both figures were entirely generated by us using publicly available datasets from the Spatial Data Repository, The Demographic and Health Surveys Program (ICF International), funded by the United States Agency for International Development (USAID). The data was accessed from [spatialdata.dhsprogram.com](https://spatialdata.dhsprogram.com) on December 4, 2024, as cited in the figure captions.

Since we created these figures, they are original works and do not infringe on any third-party copyright. We hereby confirm that we are the copyright holder of these figures and grant permission for their publication under the Creative Commons Attribution License (CC BY 4.0). We have ensured that the figure captions include the appropriate attribution and licensing information.

"Source: Spatial Data Repository, The Demographic and Health Surveys Program, ICF International. Funded by the United States Agency for International Development (USAID). Available from spatialdata.dhsprogram.com. [Accessed 04 December 2024]. Created by the author and published under the CC BY 4.0 license."

Reviewer #1: ### 1. Measuring Outcome Variables

How do you measure your outcome variable? Have you employed the same

measurement for both spatial and multilevel analyses?

Response:

We appreciate the reviewer's insightful question concerning the measurement of our outcome variable and its application in both spatial and multilevel analyses. Our study measure intimate partner violence (IPV) among married women utilizing the standardized Domestic Violence Module from the 2019 Sierra Leone Demographic and Health Survey (DHS).

This module collects self-reported data on various forms of IPV, including physical, emotional, and sexual violence, experienced by women aged 15–49 who are currently or previously married or in a union. The IPV variable is typically constructed by aggregating responses to specific questions regarding acts such as being slapped, pushed, hit, kicked, forced into sexual acts, or insulted by a partner. The responses are coded into a binary composite measure (0 = no IPV, 1 = any IPV), in line with established DHS methodology and prior studies (Seidu, Mossie). This coding indicates whether a woman has experienced any form of intimate partner violence (IPV) within a specified timeframe, such as the past 12 months or at any point in her life. This approach allows researchers to quantify and analyze the prevalence and patterns of IPV among married women in Sierra Leone.

We employed a consistent measure of intimate partner violence (IPV) for both spatial and multilevel analyses to ensure the comparability of our findings. In the spatial analysis, we linked each of the three types of IPV—Physical, Emotional, and Sexual violence—represented as binary variables, to the GPS coordinates of DHS clusters. This approach allowed us to identify geographic patterns while accounting for spatial autocorrelation. For the multilevel analysis, we maintained the same binary outcome (IPV) but focused on differentiating between individual and community-level factors using hierarchical random effects.

### 2. Choice of Spatial Interpolation Techniques

There are numerous spatial interpolation techniques, such as Inverse Distance Weighting (IDW), Kriging, splines, and natural neighbors. Why did you choose

Bayesian spatial Kriging? What are the implications and added value of using this

model in the context of your findings?

Response:

We extend our sincere gratitude to the reviewer for their insightful inquiry regarding our selection of Bayesian spatial Kriging as the methodological framework for this study. Our choice to employ this particular spatial interpolation technique is underpinned by several theoretical and practical considerations that render it particularly suitable for investigating the dynamics of intimate partner violence (IPV) in Sierra Leone.

The Bayesian approach to spatial Kriging offers significant advantages over traditional interpolation methods, such as Inverse Distance Weighting (IDW) or frequentist Kriging, particularly in the context of complex health outcome data. Notably, empirical Bayesian Kriging (EBK) excels in mitigating variability and predicting values in regions that remain unsampled by exploiting data points from neighboring locales. This technique harnesses the principles of spatial autocorrelation to discern spatial trends within the data and to forecast the underlying dependency structures[38]. The method incorporates simulations and subsetting paradigms for parameter estimation, deriving the semivariogram from existing data points, thereby enabling the estimation of values at unobserved locations using a unified semivariogram. It is crucial to acknowledge that this approach presumes the semivariogram calculated for the interpolation zone accurately represents the true semivariogram [39].

In comparative assessments against alternative spatial interpolation techniques, Bayesian Kriging has demonstrated superior performance tailored to our specific research imperatives. EBK has exhibited enhanced accuracy through the integration of auxiliary variables together with spatial autocorrelation[40]. Furthermore, Bayesian Kriging addresses the tendency of Ordinary Kriging to produce overly optimistic prediction variances by conceptualizing model parameters as random variables, which facilitates more robust uncertainty quantification[41].

The implementation of Bayesian spatial Kriging has profoundly influenced our findings, establishing it as an invaluable tool for the exploration of patterns and risk factors associated with IPV. Numerous studies have successfully employed this methodology to identify high-risk areas in conjunction with pertinent neighborhood-level characteristics[42-45]. Moreover, this modeling approach adeptly identifies hotspots, regions characterized by uncertain estimates, and covariates that elucidate the spatial patterns of the outcome variable[43]. The Bayesian spatial modeling framework has enabled researchers to account for both fixed covariate effects and spatially structured random effects, thereby facilitating a comprehensive understanding of IPV patterns and informing targeted prevention strategies[44, 45]. We recognize that the application of Bayesian spatial techniques presents computational challenges and necessitates meticulous selection of prior distributions. To address these concerns, we undertook extensive sensitivity analyses to evaluate the influence of varying prior specifications on our study's findings.

### 3. Geographic Boundaries and Units of Analysis

What geographic boundaries or unit levels were used in each analysis? Based on

the DHS data, these could be at the level of countries, regions (comprising multiple

countries) within Sub-Saharan Africa (SSA), census clusters, or GPS points of

households. It is crucial to clarify this for accurate interpretation of the spatial

analysis.

Response:

We greatly appreciate the reviewer's question regarding the geographic units used in our analyses. Clarifying these boundaries is indeed crucial for accurate interpretation of both our spatial and multilevel findings. In our study, we used sub-national regional boundaries for the SLDHS, specifically we used administrative level 2 that comprises of 16 districts to maximize analytical precision while maintaining relevance for policy implementation in Sierra Leone.

Your findings indicated a district-level analysis. How did you define districts?

Specifically, how did you identify the number of districts, and you mentioned 18

districts—how did you determine this? How many districts were included in the

study?

Response:

We thank the reviewer for this question regarding our district-level analysis. We appreciate the opportunity to clarify how we defined and incorporated districts into our study. In our analysis, we used Sierra Leone’s administrative boundaries at Level 2 (districts) as defined in the 2015 Population and Housing Census (Statistics Sierra Leone, 2016) and aligned with the 2019 Sierra Leone Demographic and Health Survey (SLDHS) geographic coding. Additionally, we included the 16 districts, which served as the primary sub-national administrative units for policy implementation and health planning.

You utilized DHS data, which is derived from a spatial repository that includes

enumeration areas for specific countries. How did this data translate into district-level

analysis? Please provide a clear description of the level of analysis used.

Response:

We appreciate the reviewer's question regarding how we translated the DHS spatial data into district-level analysis. This is a crucial methodological consideration, and we added a detailed explanation of our approach in the main manuscript from lines 109 to 124 of page 7.

### 4. Managing Missing Data

How do you manage missing data, including missing attributes and missing

coordinate data?

Response:

We thank you for your comment. We addressed missing data through multiple imputations for non-essential missing values and implemented complete case analysis when the amount of missing data was minimal. Furthermore, we utilized DHS sample weights to adjust for non-response, ensuring that our results were representative. Finally, we acknowledge the DHS program's geographic displacement protocol which occasionally results in missing or unreliable coordinates for sensitive locations, we therefore appended the IPV-related dataset with the Global Positioning System (GPS) coordinates of SLDHS and then conducted sensitivity analyses comparing results with and without append GIS.

### 5. Study Variables

Please elaborate on your outcome variable. Have you categorized it into binary

responses (yes or no)? Clearly indicate this.

Response:

We appreciate the reviewer's question regarding the outcome varaible. We made a clear explanation of the outcome varaible in the main manuscript from lines 127 to 140 of pages 7 and 8.

### 6. Spatial Autocorrelation

Which spatial matrix did you use to report spatial autocorrelation (e.g., Moran's I)?

Please articulate the implications based on your findings.

Response:

We thank the reviewers for the spatail amtrix use to report the spatil autocorrelation. In our study, to assess the spatial pattern of IPV across Sierra Leone we conducted a spatial autocorrelation analysis using Global Moran's I with an inverse-distance spatial weights matrix. This approach allowed us to evaluate whether IPV prevalence exhibited clustering, dispersion, or random distribution across districts. The inverse-distance matrix was selected because it assumes that nearby districts exert greater influence on each other than those farther apart—a theoretically plausible assumption for social phenomena like IPV, where cultural norms and resource availability often follow geographic gradients.

### 7. Multi-Level Analysis

You conducted Intraclass Correlation Coefficient (ICC) analysis. Given that your

analysis is multi-level, it would be beneficial to include Measures of Relative Risk

(MOR) and Proportional Change in Variance (PCV). What were your community-

level and individual-level variables?

Response: This study included fourteen independent variables selected based on their availability in the dataset and insights from prior research. Individual-level variables comprised the respondent's age, educational level, employment status, exposure to family planning through television, radio, and newspapers, partner's controlling behaviors, justification of spousal abuse, interparental violence, and husband's alcohol consumption. Community-level variables included gender of the ho

---

## [Editor Report · Decision Letter 1]

13 May 2025

Spatial Mapping and Determinants of Intimate Partner Violence Among Married Women In Sierra Leone

PONE-D-25-03253R1

Dear Dr. Augustus Osborne

We’re pleased to inform you that your manuscript has been judged scientifically suitable for publication and will be formally accepted for publication once it meets all outstanding technical requirements.

Kind regards,

Menen Tsegaw Biadiglgn, MPH

Academic Editor

PLOS ONE

Additional Editor Comments (optional):

Dear Dr. Augustus Osborne

I would like to thank you for addressing the comments. The manuscript has undergone a revision and extensively improved. I felt that the quality and clarity of the language, rationale, methods and reporting were of a high enough standard for us to publish the manuscript in PLOS One journal. I would like to thank you for considering PLOS One for the publication of your research. I hope the outcome of this specific submission will encourage you to submit future manuscripts to our Journal. Thus, I am thrilled to express my recommendation to accept this manuscript for publication due to my strong believe in its high quality and greater impact.

---

## [Editor Report · Acceptance letter]

PONE-D-25-03253R1

PLOS ONE

Dear Dr. Osborne,

I'm pleased to inform you that your manuscript has been deemed suitable for publication in PLOS ONE. Congratulations! Your manuscript is now being handed over to our production team.

Kind regards,

on behalf of

Ms Menen Tsegaw Biadiglgn

Academic Editor

PLOS ONE